# Interactive Video-Based Passive Drinking and Forced Drinking Education to Reduce Intention to Drink in Adolescents: A Pre-Post Intervention Study

**DOI:** 10.3390/ijerph20010332

**Published:** 2022-12-26

**Authors:** Siu Long Chau, Yongda Wu, Man Ping Wang, Sai Yin Ho

**Affiliations:** 1School of Nursing, The University of Hong Kong, Hong Kong, China; 2School of Public Health, The University of Hong Kong, Hong Kong, China

**Keywords:** adolescents, passive drinking, forced drinking, alcohol misuse, interactive video-based education, pre-post intervention study

## Abstract

Passive and forced drinking harm was prevalent but less recognized in Chinese adolescents. We educated adolescents on such harm to reduce their intention to drink. Students (*n* = 1244) from seven secondary schools in Hong Kong participated in a video-based health talk on passive and forced drinking harm. Paired *t*-test was used to assess their change in knowledge of passive and forced drinking, and health and social harm of drinking after, the health talk. McNemar’s chi-squared test and adjusted multivariable logistic regression (AOR) were used to assess their change in intention to drink and intention to quit. Students were less likely to drink (OR 0.29, 95% CI 0.19–0.42) and more likely to quit drinking (OR 3.50, 1.10–14.6) after the health talk. Increased knowledge of passive drinking was associated with less intention to drink (AOR 0.93, 0.90–0.97), increased knowledge of health harm (adjusted b 0.06, 0.05–0.08), and social harm of drinking (adjusted b 0.12, 0.10–0.16). Similar associations were observed in forced drinking (intention to drink: AOR 0.87, 0.79–0.96; health harm: adjusted b 0.16, 0.12–0.19; social harm: adjusted b 0.36, 0.28–0.43). We showed preliminary evidence that the health talk on passive and forced drinking reduced the intention to drink in adolescents.

## 1. Introduction

Harmful use of alcohol affects both developed and developing countries [1]. It contributes to 3 million deaths worldwide and is one of the leading risk factors for premature death and disability-adjusted life year loss (DALYs), in adolescents [2]. Harmful drinkers develop regular drinking habits at this stage [3,4]. In Hong Kong (HK), 66.7% of adult binge drinkers had drunk alcohol below the legal age (18 years old), and 16.6% of adult drinkers had underage drinking habits (*n* = 1087) [5]. Alcohol use is prevalent in adolescents. In a representative sample of HK secondary students (*n* = 23,288), 21.5% were current drinkers, and 7.5% binge drank in the past 12-month [3] (five or more standard drinks for males; four or more standard drinks for females on one occasion; one standard drink contains 10 g of pure alcohol) [6]. Although the prevalence of current drinking (drink at least one standard drink in the past 30-day) [4] in adolescents was lower than in western countries (e.g., 57% in European countries and 39% in the United States) [7], interventions are needed to stop the growing trend of harmful alcohol use for this target population.

The adverse effects of alcohol are not limited to drinkers themselves but also people surrounding the drinkers. Passive drinking is defined as harm resulting from others’ drinking; serious harm includes sexual harassment, unwanted intercourse, and physical assaults by the drinking person [8]. Adolescents were the primary victims of passive drinking [9]. In total, 2 in 5 local secondary students experienced passive drinking in the past 30-days [3]. The adolescence stage is life’s most critical developmental stage, and environmental factors highly shape their development [10]. Parental drinking was linked to early initiation of alcohol consumption, harmful alcohol use, delinquency, and risky behaviors (e.g., unsafe sex and drug abuse) in adolescents [11,12,13]. Costs of morbidity and death from passive drinking were high. In Australia, the morbidity of child abuse caused by drinking parents costs AUD $3.6 million annually [14].

Forced drinking is defined as drinking alcohol unwillingly, including drinking under force, playing drinking games, and drinking due to social pressure [3]. Drinking is considered socially acceptable and one of the important cultures in China [15]. Moderate alcohol consumption was perceived as beneficial to health; many parents trained their children to drink alcohol and regarded it as a behavior for success in Chinese society [16]. HK adolescents grew up in an alcohol-friendly and pro-drinking environment, with toasting and being invited to drink by peers and family members being common in secondary students [3]. Our survey found that 1 in 5 HK Chinese secondary students experienced forced drinking in life, 7.1% drank under peer pressure, and 6.3% were forced to drink by family members [3]. These activities predisposed adolescents to alcohol intoxication [17].

Cochrane systematic reviews found that school-based prevention programs were cost-effective in delaying the onset of harmful drinking in adolescents even with small effect sizes, and saved substantial government expenses on alcohol-related morbidity [18,19,20]. Yet, the content and mode of intervention delivery were still inconclusive. No study was found in PubMed and the Cochrane Database of Systematic Reviews using video-based intervention to educate students on passive drinking and forced drinking to reduce adolescent intention to drink. We aimed to raise awareness of the harm of passive and forced drinking in adolescents. We also investigated the effectiveness of interactive video-based education in reducing the intention to drink in this population.

## 2. Methods

### 2.1. Study Design

In this quasi-experimental study, we used the Education Bureau (EDB) 2018 list of secondary schools as the sampling frame. The target sample size was 1230 students from 10 public schools. We recruited local secondary schools from all five regions of Hong Kong (Hong Kong Island, Kowloon East, Kowloon West, New Territories, and New Territories West) using random sampling method and invited schools to join. Seven public schools with the same school ranking from each of the five regions participated. School participation was voluntary, with refusal mainly due to time and administrative issues. All Students in forms 1–6 (Grade 7–12), in the participating schools, were invited to the video-based health talk. Invitation letters were sent to parents via the schools to explain the health talk’s purpose and emphasize that participation was voluntary. Parents who refused to join were asked to have their children return blank questionnaires after the talk. Students’ participation was voluntary, even with parental consent. Before the health talk, students were briefed about the self-administered anonymous questionnaires, and teachers were reminded not to influence students to answer the questions. The two surveys (pre- and immediate post-questionnaire) took 5–10 min to complete, and 10 min break was given to students before completing the post-questionnaire. The completed questionnaires were inserted into opaque envelopes and sealed immediately after completion. Students who completed both questionnaires (*n* = 1244) were analyzed. Ethical approval was granted by the Institutional Review Board of the University of Hong Kong/Hospital Authority Hong Kong West Cluster (Ref. UW 19-400).

### 2.2. Intervention

The interactive video-based health talk lasted for 45 to 60 min. It was divided into three sessions, which covered the topics of the harm of drinking, knowledge of passive and forced drinking, and strategies to reject drinking invitations. In the first session, students were asked to vote for the answers regarding the myths about drinking (e.g., whether drinking a small amount of red wine daily minimizes cardiovascular disease risk). Correct answers were provided and explained after each vote, and small gifts were given to students who voted for the correct answers. In the second session, students were introduced to passive drinking and forced drinking concepts, including the harm, situations, risk factors, and strategies to avoid those harm. We created two age-appropriate cartoon animations related to the knowledge of passive and forced drinking. The videos educated students on the harm of passive drinking (a story about a drunk driving victim) and strategies to avoid forced drinking (ways to avoid drinking invitations from peers). Each video lasted 2 min, and were shown in the health talk to enhance students’ impressions. In the final session, students were taught the strategies to reject peers’ or family members’ drinking invitations (e.g., making excuses or saying no to alcohol without hesitation). Important points of the above information were summarized at the end of the health talk. 

### 2.3. Measurements

Socio-demographic characteristics and drinking behaviors were collected at baseline. Students were asked if they agreed on the following 12-items of harm of passive drinking in the pre- and post-questionnaires, including noise; sleep/study interrupted; emotionally hurt; feeling neglected; taking care of a drunk; verbal insulted; physical assault; sexual harassment; unwanted sex; financial loss; accident; and property damaged. Situations of forced drinking (5-items: asked by senior to drink; propose a toast to senior; drink under peer pressure; drink for others; and playing drinking games), health harm of drinking (5-items: drinking: would affect health; would cause cancer, is addictive; would increase blood pressure; and would cause weight gain), and social harm of drinking (9-items: drinking: would relieve stress; would induce happiness; would lead to poor academic performance; would increase one’ maturity; would reduce next drunkenness; would increase own attractiveness; would help to make new friends; is appropriate for secondary students; and benefits outweigh the risks) were assessed similarly. Intention to quit was measured by asking, “Do you want to stop drinking now?” (yes vs. no), and intention to drink was measured by asking, “Would you drink if your best friend offered you a drink?” (yes vs. no). Both questions were assessed in the pre-and post-questionnaires. The satisfaction level and perceived effectiveness of the Interactive video-based education were measured in the post-questionnaire. 

### 2.4. Statistical Analysis 

A total of 127 students (9.3%) with missing answers for over half of the questionnaire were excluded. Paired *t*-test was used to measure the change in students’ knowledge of passive drinking (total score: 12), forced drinking (total score: 5), health harm of drinking (total score: 5), and social harm of drinking (total score: 9), after the health talk. McNemar’s chi-squared test was used to assess the change of intention to quit (yes vs. no) and intention to drink (yes vs. no) after the health talk. Sensitivity analyses were conducted for the intention to quit and intention to drink outcomes (multiple imputations by chained equations to impute missing data, with inferences drawn from 50 imputed datasets). The associations of knowledge of passive drinking and forced drinking, with knowledge of drinking harm, intention to drink, and intention to quit, were analyzed using multiple linear regression (adjusted unstandardized coefficient, b) and logistic regression (adjusted odds ratio (AOR)), adjusting for sex, school grade levels, perceived affluence, and corresponding pre-talk score. The associations of socio-demographic characteristics with post-talk scores of passive drinking and forced drinking were analyzed using multiple linear regression, mutually adjusting for baseline socio-demographic characteristics and pre-talk scores. Stata 15.1 was used for all analyses. A two-tailed *p*-value of less than 5% was considered statistically significant. 

## 3. Results

Table 1 shows that 57.9% of students were female, 81.6% were middle school grade level, and 58.5% perceived their family affluence as average. A total of 63.6% never had drunk alcohol, and 16.8% drank at least monthly. 

Table 2 shows a significant increase in the scores of students’ knowledge of passive drinking (total score 12: 4.78 vs. 5.83, *p* < 0.001), forced drinking (total score 5: 2.27 vs. 2.72, *p* < 0.001), health harm of drinking (total score 5: 3.13 vs. 3.58, *p* < 0.001), and social harm of drinking (total score 9: 4.36 vs. 5.43, *p* < 0.001), after the health talk. The odds ratio from McNemar’s chi-squared test shows that students were more likely to quit (OR 3.50, 95% CI 1.10–14.6) and less likely to drink alcohol (OR 0.29, 0.19 to 0.42) after the health talk. The differences in the intention to quit (OR 1.55, 1.20 to 2.01) and intention to drink (OR 0.83, 0.74 to 0.93) outcomes, based on the multiple imputed data analyses, were significant.

Table 3 shows that an increase in knowledge of passive drinking (adjusted b 0.06, 95% CI 0.05 to 0.08) and forced drinking (adjusted b 0.16, 0.12 to 0.19), after the health talk, was associated with an increase in knowledge of the health harms of drinking. Similar positive associations were observed for knowledge of social harm of drinking (passive drinking: adjusted b 0.12, 0.10 to 0.16; forced drinking: adjusted b 0.36, 0.28 to 0.43). Students with increased knowledge of passive drinking (AOR 0.93, 95% CI 0.90 to 0.97) and forced drinking (AOR 0.87, 0.79 to 0.96) were less likely to drink after the health talk. 

Table 4 shows that students who were female (adjusted b 0.70, 95% CI 0.23 to 1.18) had a significant increase in knowledge of passive drinking after the health talk, adjusting for pre-talk scores and potential covariates. Similar positive associations were observed for forced drinking (female: adjusted b 0.28, 0.07–0.49). Students from form 6 had a significant decrease in knowledge of passive drinking (adjusted b −4.25, −7.30 to −1.20) and forced drinking (adjusted b −1.53, −2.87 to −0.20), after the health talk. 

Table 5 shows that 47.8% of students were satisfied with the health talk, with 78.7% agreeing that it heightened their knowledge of passive and forced drinking, and 61.6% agreeing that it facilitated discussion about drinking problems with parents.

## 4. Discussion

This is the first quasi-experimental study to use interactive video-based intervention to educate adolescents on the harm of passive and forced drinking, to reduce their intention to drink and promote quitting. Our previous survey found that harm was prevalent in Chinese adolescents; 38.9% and 21.9% of secondary students having experienced passive and forced drinking, respectively [3,21]. For over a decade, adolescents suffered from alcohol-related assaults from others’ drinking, and a substantial proportion of them were hospitalized, due to physical abuse by drinking parents and alcohol intoxication from peers’ forced drinking [13,17]. Our findings showed that the overall knowledge of drinking harm, passive drinking, and forced drinking increased compared to baseline; they also reported less intention to drink after the health talk. The results provide preliminary evidence that the novel interactive video-based education is a feasible method to empower students’ capability to avoid those harms, and promote their motivation to quit drinking. 

Traditional school-based alcohol prevention programs showed no significant effects on increasing the knowledge of drinking harm in adolescents [20]. Our study provided initial evidence that educating students on the harm of passive and forced drinking was associated with increased knowledge of the health and social harm of drinking. According to the Health Belief Model, increasing the perceived harm of passive and forced drinking may increase the perceived severity of alcohol use, and de-normalize drinking behaviors in adolescents [22]. Our study also showed that increased knowledge of forced drinking was associated with less intention to drink. Forced drinking (e.g., playing drinking games or being invited to drink by peers) was linked to increased alcohol consumption. These drinking activities provide opportunities for learning drinking behaviors from peers [18,23]; drinking with peers reinforces the perceived positive social reward of drinking and promotes motivation to drink [18]. In addition, playing drinking games was identified as a crucial predictor of harmful drinking in adolescents [24]. A study found that drinking game players were more likely to binge drink and engage in heavy episodic drinking on a drinking occasion than non-game players [24]. Adolescents who participated in drinking games during secondary school were associated with a higher risk of alcohol abuse in university [25]. Our results provide evidence that educating adolescents about the harm of forced drinking was crucial to de-normalize drinking behaviors and prevent them from developing risky drinking practices.

We found that the intervention increased adolescents’ knowledge of passive and forced drinking, except for older students (Grade 12) and those who perceived themselves as affluent. Older students were more likely to drink and resort to alcohol to cope with stress and anxiety as they had more external stressors (e.g., academic stress and social obligations) than younger students [26]. In addition, older students were exposed to more pro-drinking behaviors and alcohol advertisements on social media; alcohol industries advocated the benefits of drinking on these platforms and advertised that people from higher social classes would drink alcohol (e.g., red wine and whiskey) for leisure and uplift their social status [27]. These risk factors reinforced the positive social rewards of drinking. Adolescents from a higher socioeconomic status (SES) also had more alcohol misuse problems [28]. Students with higher perceived family affluence were more likely to hear parents speak of the benefits of drinking and pouring alcohol for parents [29]. Parents with higher SES were reported more as red wine drinkers and encouraged their children to try alcohol for health [29]. Our results provide support for persuading parents with higher SES not to drink in front of children and not to advocate the benefits of drinking to them.

## 5. Limitations

This study had some limitations. First, the intention to drink and intention to quit were assessed immediately after the health talk, but long-term follow-up data is needed to confirm the effects. The socially desirable answers cannot be excluded. It was less likely given that drinking is prevalent and anonymous questionnaires were used. Second, the causal effects between the intervention and outcomes cannot be concluded, and confounding bias might exist due to the lack of randomization. We have minimized its effects by adjusting for potential confounding variables, and a randomized controlled trial is needed to provide more robust evidence. Third, a students’ experience taking the pre-test questionnaire might affect their performance on the post-test questionnaire (testing effect). We used a 10 min washout period after the health talk to minimize its effect on students’ performance on the post-test questionnaire.

## 6. Conclusions

We found that our interactive video-based education increased students’ overall knowledge of drinking harm; they also reported less intention to drink after the intervention. Increased knowledge of passive and forced drinking were associated with increased knowledge of drinking harm and less intention to drink. Students who were older and perceived themselves as affluent were not receptive to our intervention; more intensive intervention is needed for these specific groups. This study provides implications that educational programs at school, such as teaching students about the knowledge of passive and forced drinking, can be a feasible strategy to prevent students from underage drinking and developing alcohol misuse problems in the future. 

## Figures and Tables

**Table 1 ijerph-20-00332-t001:** Socio-demographic characteristics and drinking behaviors of 1244 participants.

	*n* ^a.^	%	(95% CI)
Sex			
Male	513	42.1	(39.4–44.9)
Female	705	57.9	(55.1–60.6)
School grade level ^b.^			
Middle school (F.1–3)	1010	81.6	(79.3–83.6)
High school (F.4–6)	228	18.4	(16.4–20.7)
Perceived family affluence			
Below average	305	24.8	(22.5–27.3)
Average	719	58.5	(55.6–61.2)
Above average	206	16.7	(14.8–18.9)
Drinking frequency			
Never	752	63.6	(61.0–66.3)
Yearly or less	232	19.6	(17.4–22.0)
Monthly or less	103	8.7	(7.2–10.5)
1–3 times per month	64	5.4	(4.3–6.9)
1–6 times per week	19	1.6	(1.0–2.5)
Everyday	13	1.1	(0.6–1.9)

^a^. Observation (*n*) is not 1244 due to nonresponse. ^b^. Forms 1–6 are equivalent to Grades 7–12 in the North American-based system.

**Table 2 ijerph-20-00332-t002:** Change in knowledge of passive drinking, forced drinking, drinking harm, intention to quit, and intention to drink after the health talk in 1244 participants ^a^.

				McNemar’s Chi-Squared Test	Multiply Imputed Data Analyses
	Pre Mean (SD)/	Post Mean (SD)/	*p*-Value ^b.^	OR (95% CI) ^c.^	OR (95% CI) ^c.^
	*n* (%)	*n* (%)			
Passive drinking (*n* = 1244)					
Harms (total score = 12)	4.78 (3.87)	5.83 (4.57)	<0.001		
Forced drinking (*n* = 1244)					
Situations (total score = 5)	2.27 (1.64)	2.72 (2.00)	<0.001		
Knowledge of drinking harm (*n* = 1244)					
Health harm (total score = 5)	3.13 (1.23)	3.58 (1.42)	<0.001		
Social harm (total score = 9)	4.36 (2.89)	5.43 (3.23)	<0.001		
Intention to quit (*n* = 127)	37/127 (29.1)	47/127 (37.0)	0.018	3.50 (1.10, 14.6) *	1.55 (1.20, 2.01) *
Intention to drink (*n* = 942) ^d.^	472/942 (50.1)	383/942 (40.7)	<0.001	0.29 (0.19, 0.42) ***	0.83 (0.74, 0.93) **

^a^. Observation (*n*) only included consistent answers to drinking frequency questions and excluded students who never drink and stop drinking. ^b^. *p*-value was calculated by McNemar’s chi-squared test and paired *t*-test. ^c^. The odds ratio was calculated by McNemar’s chi-squared test. ^d^. Observation (*n*) is not 1244 due to nonresponse. * *p* < 0.05; ** *p* < 0.01; and *** *p* < 0.001.

**Table 3 ijerph-20-00332-t003:** The association of knowledge of passive drinking and forced drinking with knowledge of health harm and social harm of drinking, intention to drink, and intention to quit in 1244 participants.

Variables	b (95% CI)	*p*-Value ^b.^
	/OR (95% CI) ^a.^	
Knowledge of health harm (*n* = 1209; total score = 5)		
Passive drinking (post-score)	0.06 (0.05, 0.08)	<0.001
Forced drinking (post-score)	0.16 (0.12, 0.19)	<0.001
Knowledge of social harm (*n* = 1209; total score = 9)		
Passive drinking (post-score)	0.12 (0.10, 0.16)	<0.001
Forced drinking (post-score)	0.36 (0.28, 0.43)	<0.001
Intention to drink (*n* = 1209; Yes vs. No)		
Passive drinking (post-score)	0.93 (0.90, 0.97)	0.001
Forced drinking (post-score)	0.87 (0.79, 0.96)	0.004
Intention to quit (*n* = 1203; Yes vs. No)		
Passive drinking (post-score)	1.02 (0.95, 1.09)	0.593
Forced drinking (post-score)	1.04 (0.89, 1.22)	0.606

^a^. Regression was adjusted by sex, school grade level, perceived affluence, and pre-talk score. ^b^. *p*-value was calculated by multiple linear regression and logistic regression.

**Table 4 ijerph-20-00332-t004:** The association of socio-demographic characteristics with the post-talk score of passive drinking and forced drinking in 1244 participants.

Variable	Pre Mean (SD)	Post Mean (sd)	*p*-Value ^a.^	b (95% CI) ^b.^	*p*-Value ^c.^
Passive drinking (*n* = 1209)					
Harm (total score = 12)					
Sex					
Male	4.83 (4.10)	5.48 (4.70)	0.001	REF	
Female	4.79 (3.69)	6.16 (4.46)	<0.001	0.70 (0.23, 1.18)	0.004
School grade level ^d.^					
Form 1	4.52 (3.79)	5.72 (4.56)	<0.001	REF	
Form 2	5.01 (3.98)	5.62 (4.40)	0.010	−0.45 (−1.07, 0.17)	0.158
Form 3	5.18 (4.03)	6.53 (4.72)	<0.001	0.28 (−0.34, 0.90)	0.374
Form 4	4.11 (3.26)	5.52 (4.54)	<0.001	−0.28 (−1.08, 0.52)	0.494
Form 5	4.46 (3.67)	5.43 (4.34)	0.009	−0.25 (1.26, 0.75)	0.619
Form 6	5.71 (5.22)	2.00 (4.47)	0.059	−4.25 (−7.30, −1.20)	0.006
Perceived family affluence					
Affluent	5.21 (4.24)	4.67 (4.48)	0.518	REF	
Above average	4.97 (4.09)	5.82 (4.51)	0.015	1.10 (−0.29, 2.48)	0.120
Average	4.74 (3.76)	5.89 (4.55)	<0.001	1.20 (−0.06, 2.47)	0.063
Below average	4.40 (3.66)	5.80 (4.56)	<0.001	1.30 (−0.05, 2.65)	0.059
Poor	5.67 (4.45)	6.36 (4.83)	0.183	1.40 (−0.07. 2.87)	0.062
Forced drinking (*n* = 1209)					
Situation (total score = 5)					
Sex					
Male	2.04 (1.68)	2.45 (2.08)	<0.001	REF	
Female	2.45 (1.58)	2.96 (1.91)	<0.001	0.28 (0.07, 0.49)	0.010
School grade level ^d.^					
Form 1	2.21 (1.72)	2.75 (2.04)	<0.001	REF	
Form 2	2.29 (1.67)	2.64 (1.92)	<0.001	−0.19 (−0.46, 0.08)	0.170
Form 3	2.36 (1.54)	2.84 (2.01)	<0.001	−0.05 (−0.32, 0.21)	0.693
Form 4	2.18 (1.55)	2.71 (1.99)	<0.001	−0.15 (−0.50, 0.20)	0.406
Form 5	2.29 (1.63)	2.70 (2.00)	0.041	−0.09 (−0.53, 0.35)	0.699
Form 6	2.28 (2.06)	1.14 (2.03)	0.231	−1.53 (−2.87, −0.20)	0.025
Perceived family affluence					
Affluent	2.74 (1.69)	2.40 (1.99)	0.236	REF	
Above average	2.13 (1.68)	2.71 (1.96)	<0.001	0.52 (−0.08, 1.13)	0.091
Average	2.33 (1.60)	2.79 (1.99)	<0.001	0.48 (−0.07, 1.04)	0.087
Below average	2.09 (1.59)	2.65 (1.97)	<0.001	0.48 (−0.11, 1.07)	0.112
Poor	2.38 (1.84)	2.61 (2.18)	0.272	0.36 (−0.28, 1.00)	0.266

^a^. *p*-value was calculated by paired t-test. ^b^. All variables were mutually adjusted and adjusted for the pre-talk score. ^c^. *p*-value was calculated by multiple linear regression. ^d^. Forms 1–6 are equivalent to Grades 7–12 in the North American-based system.

**Table 5 ijerph-20-00332-t005:** Overall satisfaction levels and perceived effectiveness of the health talk in 1244 participants.

	*n* ^a.^	% (95% CI)
Do you satisfy with the workshop? (*n* = 1228)		
Not satisfy	205	16.7 (14.7–18.9)
Neutral	436	35.5 (32.9–38.2)
Very satisfy	587	47.8 (45.0–50.6)
Do you agree that the workshop helped broaden your knowledge of passive and forced drinking? (*n* = 1225)		
Disagree	261	21.3 (19.1–23.7)
Agree	964	78.7 (76.3–80.9)
Do you agree that the workshop facilitates discussing the drinking problem with your parents? (*n* = 1224)		
Disagree	470	38.4 (35.7–41.2)
Agree	754	61.6 (58.8–64.3)

^a^. Observation (*n*) is not 1244 due to nonresponse.

## Data Availability

The dataset generated during and/or analyzed during the current study are available from the corresponding author on reasonable request.

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
