# Peer review of "Interactive Video-Based Passive Drinking and Forced Drinking Education to Reduce Intention to Drink in Adolescents: A Pre-Post Intervention Study"

_ijerph, 2022, doi:10.3390/ijerph20010332_

Round 1
Reviewer 1 Report
This is a quasi-experimental experiment with promising results. As the authors note in the limitations section, there are several biases that may confound the results. The timing of the post-test was so soon after the pre-test and video viewing that I am not sure the results are valid. Is there any way to conduct a follow-up survey, or were there questions asked on the that were not related to the video that could be used as a sensitivity/validity check on the changes in the outcomes assessed? For example, if knowledge about healthy diets also improved, or an aspect of alcohol use that was not covered in the video that would be a sign that the video had the intended effect, and it wasn't social desirability or testing effect.
Reviewer 2 Report
In this study, the authors conducted a video-based education program related to alcohol use among secondary school students in Hong Kong; pre- and post-tests and questionnaires were used to assess the changes in knowledges and perceptions of the participants. This might be an interesting topic; however, there are several critical insufficiencies in this manuscript must be addressed.
#1 Overall
My first concern is the representativeness of the subjects included in the study. As mentioned in the Methods section, only 7 out of all 479 schools (1.5%) were included in the study. Accordingly, selection bias is inevitable. Were the 7 schools private or public? Were they located in a specific area? Or were their basic profiles (smoking rate, drinking rate…and so on) very different from the others? Information on their characteristics should be clarified.
#2 The problem of missing values in this study was not addressed. Subjects of study are different among analyses because of missing values in different variables. For example, the number of “intention of quit” was only 127; the number of “intention of drink” was 942, and some others.
#3 Education on harm or risk of alcohol use among adolescents is quite common in many countries. Was the video-based program a standard or recommended one? Or what was the difference from others?
#4 It is not clear why the associations of knowledge of passive drinking and forced drinking with knowledge of health harm and social harm of drinking (shown in Table 3) are worthy to be discussed in this study. Are they with a specific interpretation?
#5 Please justify the number of digits in p-values shown in Tables. It is not common to show “<0.05” or “>0.05” but their exact values in an epidemiological paper.
#6 In this study, “intention to drink” was assessed at the same time (questionnaire) with knowledge of harms of alcohol. Naturally, the intention was very likely to be part of the “knowledge.” Could it be explained as “likely to drink…” as written in Line 146 and 159 and some others? The results might be overinterpreted.
#7 Line 210
“Older students were more like….”
>>> Many 12 Graders are older than 18 years (legal drinking age in Hong Kong). It is not clear why the authors used this group as reference in the analysis shown in Table 4.
#8 Line 228-230
Is lack of randomization an issue for this study design? What is the possible study design for this study if the randomized controlled trial is available?
